# Reversal of Postnatal Brain Astrocytes and Ependymal Cells towards a Progenitor Phenotype in Culture

**DOI:** 10.3390/cells13080668

**Published:** 2024-04-12

**Authors:** Dimitrios Kakogiannis, Michaela Kourla, Dimitrios Dimitrakopoulos, Ilias Kazanis

**Affiliations:** 1Lab of Developmental Biology, Department of Biology, University of Patras, 26504 Patras, Greece; dimitriskak97@gmail.com (D.K.); michaelakou@nurs.uoa.gr (M.K.); dimitris.dimitrakopoulos@unibas.ch (D.D.); 2Institute of Physiological Chemistry, University Medical Center, Johannes Gutenberg University, 55099 Mainz, Germany; 3Biology-Biochemistry Lab, Faculty of Nursing, School of Health Sciences, National and Kapodistrian University of Athens, 11527 Athens, Greece; 4Biozentrum, University of Basel, 4056 Basel, Switzerland; 5School of Life Sciences, University of Westminster, London W1W 6UW, UK

**Keywords:** neural stem cells, astrocytes, ependymal cells, mitosis, quiescence, cell cultures, reprogramming, differentiation

## Abstract

Astrocytes and ependymal cells have been reported to be able to switch from a mature cell identity towards that of a neural stem/progenitor cell. Astrocytes are widely scattered in the brain where they exert multiple functions and are routinely targeted for in vitro and in vivo reprogramming. Ependymal cells serve more specialized functions, lining the ventricles and the central canal, and are multiciliated, epithelial-like cells that, in the spinal cord, act as bi-potent progenitors in response to injury. Here, we isolate or generate ependymal cells and post-mitotic astrocytes, respectively, from the lateral ventricles of the mouse brain and we investigate their capacity to reverse towards a progenitor-like identity in culture. Inhibition of the GSK3 and TGFβ pathways facilitates the switch of mature astrocytes to Sox2-expressing, mitotic cells that generate oligodendrocytes. Although this medium allows for the expansion of quiescent NSCs, isolated from live rats by “milking of the brain”, it does not fully reverse astrocytes towards the bona fide NSC identity; this is a failure correlated with a concomitant lack of neurogenic activity. Ependymal cells could be induced to enter mitosis either via exposure to neuraminidase-dependent stress or by culturing them in the presence of FGF2 and EGF. Overall, our data confirm that astrocytes and ependymal cells retain a high capacity to reverse to a progenitor identity and set up a simple and highly controlled platform for the elucidation of the molecular mechanisms that regulate this reversal.

## 1. Introduction

Astrocytes constitute the largest glial population in the mammalian brain and they are scattered throughout the parenchyma [1]. In mice, they are the progeny of radial glial cells, generated either via direct final divisions during early postnatal life [2,3] or via embryonic, intermediate, and astroglial progenitors [4,5,6]. Historically, they were considered a relatively homogeneous population of supporting cells, even though the presence of subpopulations of different morphologies (protoplasmic versus fibrotic or even cortical layer-specific types) [5,7] and of differential marker expressions (for example, regarding GFAP or S100β) has been described [8,9]. Recently, the astrocytic heterogeneity has been confirmed at the single-cell transcription level [10,11,12,13] and it has revealed that gene expression varies significantly between astrocytes located at different areas of the brain and, surprisingly, showed common expression patterns between astrocytes and neurons of the same area [14].

There are, at least, three major indications suggesting that astrocytes retain a high plasticity capacity in vivo. Firstly, in response to injury, degeneration, or stress, they show different levels of activation, manifested with changes in shape, marker expression, and mitotic activity [15]. The astrocytes closest to the injury become highly intercalated and form the gliotic scar, a structure that encircles the area of the lesion and limits the expansion of secondary degenerative processes [16], with ambiguous effects on tissue repair and axon regeneration [17]. Secondly, the neural stem cells (NSCs) that survive in the postnatal brain, clustered within specialized microenvironments called niches, are of astroglial morphology [18]. In other words, the small pool of multipotent, self-renewing cells of the mature brain is a GFAP-expressing cell population, their existence being confirmed only by the generation of a progeny of different lineages (neurons, oligodendrocytes, and astrocytes). For example, the NSCs of the subependymal zone (SEZ) niche cannot be clearly distinguished from the other astrocytes of the niche with any definitive marker [19,20]; albeit, one morphologic difference is that they extend one monociliated process in between ependymal cells to reach the cerebrospinal fluid [21] and one longer process to contact blood vessels [18,22]. Thirdly, experimental work has shown that in response to an injury, such as a mechanical cortical lesion, a fraction of astrocytes exhibit stem/progenitor properties when isolated and cultured [23,24]. It is now established that Notch signalling is crucial for this spontaneous reprogramming [25] and accumulating reports have provided evidence that astrocytes can be converted (via targeted reprogramming) to neurons [26,27,28,29].

Ependymal cells are a glial population of specialized morphology (multiciliated, epithelial-like, and cuboid cells) that form the walls of the ventricular system of the brain and of the spinal cord [30]. Those lining the central canal have been shown to retain high plasticity potential. In response to injury, they can generate astrocytes and oligodendrocytes while they exhibit neurogenic potential in vitro [31,32]. In contrast, ependymal cells of the brain lack such potential, even though their contribution to the formation of the niche microenvironment of the SEZ is well established [33,34,35].

Based on the above, astrocytes and ependymal cells can be considered cell populations positioned closer to the neural stem/progenitor identity in a hypothetical “stem cell” to “fully mature cell” continuum, especially when compared to neurons or oligodendrocytes. The abundance of astrocytes, in combination with their still not fully known functional repertoire, as well as the functional importance of ependymal cells, in combination with their limited numbers and their inability to regenerate, make these two cell populations ideal targets for investigation. Here, we show that both these pools can be induced to reverse, to different degrees, towards a progenitor state with simple changes in their culture conditions. This confirms the continuum hypothesis and paves the way for a more targeted investigation of key molecular pathways that control the plasticity capacity of postnatal brain cells.

## 2. Materials and Methods

### 2.1. Animals

Adult male and female B6CBAC wild-type mice (1–3 months old) were used for the isolation of postnatal brain neural stem cells and ependymal cells. Adult female Wistar rats (2–4 months old) were used for the isolation of SEZ-derived cells via brain milking. Animals were kept in the animal facilities of the University of Patras, in standard laboratory polyacrylic cages (3–5 mice/cage), under a relative humidity of 50–60%, a controlled temperature (22 ± 1 °C), and a steady light–dark cycle (12 h/12 h), with free access to water and food. Their breeding and maintenance were in accordance with the European Communities Council Directive Guidelines (86/609/EEC) for the care and use of laboratory animals, as implemented in Greece by the Presidential Decree 56/2013 and approved and scrutinized by the Prefectural Animal Care and Use Committee (Protocol number: 5675/39/18-01-2021 and reference number of establishment: EL 13BIO04) and the Animal Welfare and Ethical Review Committee of the University of Patras.

### 2.2. Brain Milking

The isolation of neural stem and progenitor cells from the brains of live rats was conducted with the “brain milking” protocol described in detail in [36]. Briefly, rats were stereotaxically injected (coordinates: anterioposterior = 0.3 mm, lateral = +1.2 mm, depth = 3.5 mm) with 2 μL of a release cocktail, consisting of *Clostridium perfringens* neuraminidase (500 mU, Merck, Sigma-Aldrich, Sain Louis, MO, USA, Cat: N2876), integrin-β1-blocking antibody (1 μg, BD Pharmingen, San Diego, CA, USA, Cat: 555002), and FGF-2 (0.5 μg, Peprotech, London, UK, Cat: 100-18B). After three days, we performed four CSF liquid biopsies (“draining” of the brain) from the *cisterna magna* of the anaesthetized animals, acquiring approximately 120 μL of CSF each time, with an interval of 40 min between the collections. Biopsies were immediately mixed with an ice-cold neurosphere medium.

### 2.3. Reagents for Cell Cultures

Dulbecco’s modified Eagle’s mediumcontaining—high levels of glucose and pyruvate (DMEM, 11995-065), Neurobasal medium (21103049), DMEM-F12 (10565018), and Poly-D-Lysine (PDL, A3890401)—as well as B27 (17504-044), B27 without retinoic acid (12587010), and N2 (17502-048) culture supplements, were obtained from Thermo Fisher Scientific (Waltham, MA, USA). Fibroblast growth factor-2 (FGF-2, 100-18B) and epidermal growth factor (EGF, 315-09) were purchased from Peprotech (Londin, UK). Accutase was obtained from PAN-Biotech (P10-21100, Aidenbach, Germany). The glycogen synthase kinase-3 (GSK-3), CHIR99021 (SML1046), the TGFβ kinase/activin receptor-like kinase (ALK 5) inhibitor, A 83-01 (SML0788), and neuraminidase (N2876) were purchased from Merck, Sigma-Aldrich (Sain Louis, MO, USA).

### 2.4. Neurosphere Cultures

To culture postnatal brain neural stem and progenitor cells in the form of neurospheres, the whole lateral walls of the lateral ventricles (where the SEZs are located) of adult mice were dissected under a stereoscope, dissociated with accutase (37 °C, 20 min), and were resuspended in a standard NSC proliferation medium containing a high-glucose DMEM medium, supplemented with 2% B27 and 1% N2, as well as with FGF-2 and EGF (final concentration for both 20 ng/mL). In these conditions, NSCs are grown in the form of 3D, freely floating aggregates with a self-renewing capacity, called neurospheres, and are passaged every 5–7 days.

### 2.5. Astrocyte Generation and Culture

Astrocytes were generated by dissociating neurospheres (passages 5–10) with accutase and plating 60,000 cells/well on PDL-coated coverslips in wells of 24-well plates or 300,000 cells on PDL-coated wells of 6-well plates. Cells were cultured in DMEM, complemented with 10% heat-inactivated FBS, for 7 days, changing the medium every two days. To achieve the de-differentiation of astrocytes, the cultures’ medium was changed to either (i) a NSC-stemness medium, made of Neurobasal (49%) and DMEM-F12 (49%) media, complemented with B27^-RA^ (1%), N2 (0,5%), FGF-2 (20 ng/mL), GSK-3 inhibitor CHIR99021 (3 μM), and ALK4/5/7 or TGF-beta/smad inhibitor A83-01 (0.5 μM), or (ii) a lineage medium, made of Neurobasal (49%) and DMEM-F12 (49%) media, complemented with B27^-RA^ (1%), N2 (0,5%), and FGF-2 (20 ng/mL). Cells were kept in culture for 7 days, changing the medium every 2 days. To test the final differentiation potential of cultured astrocytes, the medium was changed to a growth-factor-free, serum-free medium made of DMEM, B27 (2%), and N2 (1%) for 5 days, changing the medium every 2 days.

### 2.6. Ependymal Cell Culture Generation and Neuraminidase Assays

Ependymal cells were obtained by dissecting the periventricular areas, around the lateral ventricles of adult mice, under a stereoscope. The tissue was dissociated with accutase (37 °C, 20 min) and cells obtained from 3 mice were pooled together and resuspended in a total of 450 μL of DMEM+ 10% heat-inactivated FBS medium. Cells were plated on 9 PDL-coated coverslips in wells of 48-well plates in 50 μL droplets and were left to adhere at room temperature for 20 min. Cells were then kept in culture with 200 μL of medium for up to 5 days without changing the media, only supplementing them to retain volume. Neuraminidase (stock solution: 20 U/mL in water) was added directly to the wells in order to create the desired final concentration.

### 2.7. Immunocytochemistry and Antibodies

Cells were fixed with 2% paraformaldehyde (PFA) (15 min at room temperature/RT) and were processed for immunofluorescence staining using standard protocols. Briefly, cells were incubated with blocking buffer (3%BSA, 0.1% Triton x-100 from Merck, Sigma-Aldrich, Sain Louis, MO, USA, in PBS) for 1 h at RT and primary antibody incubation (in blocking buffer) was performed overnight at 4 °C. The next day, samples were incubated with the appropriate secondary antibodies for 2 h at RT and were mounted with mowiol. The following antibodies were used (name, species raised in, dilution, provider, and catalogue number):

DCX (rabbit, 1/800, Abcam, Cambridge, UK, ab18723), EGFR (mouse 1/200, Abcam, ab30), GFAP (goat, 1/700, Abcam, ab53554), ID3 (mouse, 1/200, Santa Cruz Biotechnology, Dallas, TX, USA, sc-56712), Ki67 (rabbit, 1/500, Abcam, ab16667), NESTIN (chicken, 1/200, Abcam, 130417), OLIG2 (rabbit, 1/300, Millipore, Livingston, UK, AB9610), PCNT (mouse, 1/500, BD Biosciences, Wokingham, UK, 611815), Phosphorylated Histone-3 (rabbit, 1/500, Abcam, ab80612), s100β (rabbit, 1/500, Abcam, ab52642 and mouse Sigma, 1/200, S2532), SOX2 (goat, 1/200, Santa Cruz, sc-17320), TUBULIN βIII (mouse, 1/500, Abcam, ab7751), TUBULIN acetylated (mouse, 1/1500, Sigma-Aldrich, T6793), and β-catenin (mouse, 1/200, Santa Cruz Biotechnology and rabbit, 1/500, Abcam, ab16051).

Appropriate secondary antibodies conjugated with fluorescence dyes were purchased from Thermo Fisher Scientific (molecular probes, Eugene, OR, USA) and Biotium (Fremont, CA, USA) and raised in donkey or goat IgGs with fluorophores of 568 nm, 488 nm, or 647 nm.

### 2.8. Immunohistochemistry

Original images have been produced and archived from tissue used and processed as described in previous publications. For the post-brain milking tissue shown in Appendix A, see [37].

### 2.9. Quantitative Reverse Transcription Polymerase Chain Reaction

Total RNA was extracted using the QIAGEN miRNeasy Mini Kit (Germantown, MD, USA, Cat: 217004) and cDNA synthesis was conducted using the High-Capacity cDNA Reverse Transcription Kit (Applied Biosystems, San Francisco, CA, USA, Cat: 4374966), following the manufacturer’s protocols. A quantitative reverse transcription polymerase chain reaction (qRT-PCR) was performed as per the manufacturer’s instructions using the Kapa Sybr Fast Universal qPCR Kit (Sigma-Aldrich, Cat: KK4600) and results were generated with the MxPro QPCR 4.10 software. The primers (Sigma-Aldrich) used in the experiments were as follows:(1)*Rpl19* (housekeeping gene)

fp CCGACGAAAGGGTATGCTCA

rp GGGCAACAGACAAAGGCTTG

(2)
*Sox9*


fp GTACCCGCATCTGCACAAC

rp CTCCTCCACGAAGGGTCTCT

(3)
*Glast*


fp GTTCCCTGGGGAGCTTCT

rp TTACTATCTAGGGCCGCCATT

(4)
*Aldh1l1*


fp CTCGGTTTGCTGATGGGGACG

rp GCTTGAATCCTCCAAAAGGTGCGG

(5)
*Glut-1*


fp GAAGTGAAAGAGCGGGTGAG

rp CTGTTGACCAGCGCAAAG

(6)
*Tbr1*


fp CAAGGGAGCATCAAACAACA

rp GTCCTCTGTGCCATCCTCAT

(7)
*Pitx3*


fp ACCCTCCGCTTCCAGAAC

rp GAGGCCTTCTCCGAGTCAC

(8)
*Gbx2*


fp GCTGCTCGCTTTCTCTGC

rp GCTGTAATCCACATCGCTCTC

### 2.10. Imaging, Cell Counts, and Statistical Analysis

Images were taken using a Leica SP8 confocal microscope. For cell counts, at least 15 random optical fields per coverslip (10 mm diameter) were acquired with the x63 objective lens. Statistical analyses were performed using the IBM SPSS 29 statistical software and Microsoft Office Excel and the graphs were constructed in the GraphPad Prism 5.0 software. For statistical comparisons, we performed Student’s *t*-tests (for 2 groups) or one-way ANOVA, followed by post hoc tests. Probability values lower than *p* = 0.05 were considered statistically significant.

## 3. Results

### 3.1. De-Differentiation of Postnatal Brain Astrocytes via Cell Culture Media Modifications

We first obtained postnatal brain neural stem and progenitor cells (NSPCs) by dissecting and dissociating the SEZ stem cell niches of postnatal mice and growing the cells in a typical NSPC medium, which includes FGF-2 and EGF. NSPCs were maintained in the form of free-floating colonies, called neurospheres, which were passaged between five and eight times before use. Mouse astrocytes were generated by differentiating NSPCs for 7 days in DMEM complemented with 10% FBS. At the end of the 7 days, we ended up with cultures consisting of approximately 80% GFAP immunopositive cells (Figure 1A,D). Almost 50% of the cells were immunopositive for nestin (a marker of neural progenitor identity with expression carried on in the glial lineage) [38,39], the majority being double-positive for GFAP and nestin, while there were no Sox2 [40] or Ki67+ immunopositive cells, indicating that the cells had lost their progenitor identity and were post-mitotic, respectively (Figure 1D). Cells in these cultures exhibited the typical morphology of protoplasmic astrocytes (Figure 2A and Appendix A) and their molecular characterization, using RT-qPCR, revealed expression of the astrocytic markers *Sox9*, *GLAST*, *Aldh1l1*, and i(Appendix A). Based on recent experimental work showing that astrocytes, at different anatomical positions, express genes that mark local populations of neurons [14], we looked for the expression of genes, such as *Tbr1*, *Pitx3*, and *Gbx2*, that act to determine and specify cortical, nigral, and thalamic neurons, respectively, and only *Pitx3* was found to be expressed (data not shown). We subsequently switched culture conditions by growing the cells in supplemented DMEM (serum-free, growth-factor-free), which induces the final differentiation of cells, without instructing specific lineage [41] and we fixed cells after 5 days. We found a homogeneously astrocytic (GFAP+) population devoid of Olig2+ or βIII-tubulin+ (Tubb3) cells (marking cells of oligodendroglial and neuronal lineage, respectively) (Figure 2A,D; black bars). Overall, we concluded that by culturing NSPCs in 10%FBS for one week, we were able to generate cells that exhibited key properties and markers of terminally differentiated astrocytes.

To assess the de-differentiation capacity of the post-10% FBS astrocytes, we cultured them, for 7 more days, in two different media: either a NSC-lineage medium, that included only FGF-2 and promoted lineage progress, or a medium that has been previously shown to enhance “stemness” and to expand the bona fide NSC pool [42], that included inhibitors of GSK-3β and of TGF-β. Culture in the NSC-lineage medium resulted in cells with diverse morphology and an almost homogeneous co-expression of GFAP and a panel of neural progenitor identity markers, such as nestin, Sox2, and the EGF receptor (EGFR) [43] (Figure 1B,D,E,G; dark grey bars and Appendix A). Culture in the NSC-stemness medium led to cells acquiring a homogeneous, fibrous morphology (Appendix A), as well as to a strikingly uniform expression of Sox2 and a strong increase in the presence of nestin and EGFR immunopositive cells. The most notable differences to the effects of the NSC-lineage medium were that (i) nestin+ cells were significantly fewer while EGFR+ cells were significantly more; (ii) the percentage of GFAP+ cells did not change compared to the starting 10% FBS population; and (iii) a low number of Ki67+ cells emerged (Figure 1C,D,F,G). We also examined the presence of ID3, a marker of quiescent NSCs, and it was absent from cells cultured in either media (Appendix A).

Any reversal towards a de-differentiated, progenitor-like identity should be accompanied by the re-emergence of multipotency; thus, after the 7-day de-differentiation step, (NSC-lineage or -stemness media) cells were cultured in supplemented (serum and growth-factor free) DMEM for another 5 days. As expected, astrocytes were present after the differentiation of both the NSC-lineage and the NSC-stemness cell populations, albeit at significantly lower percentages when compared to the differentiated post-10% FBS cells and with NSC-stemness cells exhibiting the lowest astrogliogenic potential (Figure 2B,D). Importantly, both cell populations re-acquired a (similar) oligodendrogenic potential but failed to generate neurons.

Overall, our data revealed that the most efficient de-differentiation of a population of post-mitotic, Sox2 immunonegative, and uniformly astrogliogenic cells was achieved using the NSC-stemness medium that transformed them into cells with low astrogliogenic and high oligodendrogenic potential, with uniform re-expression of Sox2 and with re-emerging mitotic activity. Nevertheless, these significantly de-differentiated cells did not show neurogenic activity and did not include any quiescent NSCs.

### 3.2. Quiescence Can Be Promoted in Postnatal Brain, SEZ-Derived, Astroglia-Like NSPCs in Culture

In order to assess if the failure to reverse the astrocytes up to the level of ID3 expression, which would be compatible with an endogenous quiescent NSC identity, could be due to the inability of the NSC-stemness medium to foster a bona fide NSC identity, we tested this medium on rat NSPCs obtained by brain milking. This method allows the direct isolation of NSPCs from the brains of live adult rats, without the use of aggressive, tissue-dissociation protocols, thus allowing the cells to retain in higher fidelity their endogenous properties [36,37]. We confirmed the presence of cells phenotypically compatible with endogenous, quiescent NSCs (GFAP+, ID3+, EGFR-) in the samples obtained after milking (Figure 3A). When these cells were cultured for 7 days in the typical neurosphere-growing medium (in the presence of EGF and FGF2), expression of ID3 was observed in approximately 40% of the cells, EGFR started to be expressed, and a significant fraction of cells expressed Dcx, a marker of neuronal commitment (Figure 3B–I). When milking-derived cells were cultured in the NSC-lineage medium, the expression of ID3 remained at similar levels, with the percentage of nestin+ progenitor cells becoming significantly reduced and the percentage of EGFR+ cells becoming significantly increased. Neuronal commitment remained at similar levels. On the other hand, culture in the NSC-stemness medium resulted in a significant increase in the percentage of ID3+ and of nestin+ cells, as well as a significant reduction in neuronal commitment (Figure 3A–I). Mitotic activity remained at similar levels irrespective of the culture medium, with approximately 50% of the cells expressing Ki67 (data not shown). Overall, our data confirmed that the behaviour of postnatal brain NSCs, which are of the astroglial phenotype, can be significantly manipulated in vitro, with the NSC-stemness medium specifically expanding the ID3+ pool (uncommitted NSCs) and restricting the pool of EGFR+ and Dcx+ cells.

### 3.3. Postnatal Brain-Derived Ependymal Cells Regain Mitotic Activity in Culture, under Stress, and in the Presence of Growth Factors

Ependymal cells are ciliated cells that form the walls of the ventricular system. Limited ependymal loss, specifically in the SEZ area, can be restored by NSCs of the niche [44]; however, larger ependyma denudations result in gliosis and hydrocephalus [30]. Thus, the consensus is that the brain’s ependymal cells lack mitotic and regenerative capacity. However, the ependymal cells that line the spinal cord’s central canal retain the capacity to proliferate [32], indicating that the ependymal fate is not dominantly restrictive for proliferation. Therefore, we investigated if postnatal mouse brain ependymal cells can exhibit mitotic activity in vitro. Mixed cell cultures, which included approximately 25% healthy ependymal cells, could be prepared only by dissecting and dissociating periventricular areas of the lateral ventricles that were subsequently cultured in DMEM complemented with 10% FBS. Ependymal cells were identified by the combined expression of at least two of the following markers: S100β (a marker of astrocytes and ependymal cells) [37], β-catenin [22], pericentrin (PCNT, a protein that marks basal bodies, for which we did not quantify the staining but followed an on/off approach), and acetylated α-tubulin (to mark directly the cilia) (Figure 4A–C). The cells were kept in culture for 5 days, with the percentage of ependymal cells resting stable (Figure 4D). The majority of ependymal cells remained in, or formed, clusters (Figure 4D) and we could identify three major subtypes: (a) ependyma with large, typically cuboid cytoplasm; (b) ependymal cells of small, again, typically cuboid cytoplasm; (c) ependymal cells without α-tub+ cilia, with an elongated shape and extending processes (Figure 4E,F). The presence of the last ependymal pool decreased over time in the culture. Finally, we could also detect GFAP+ astrocytes, with a subpopulation co-expressing β-catenin but no other ependymal markers. The presence of these cells increased over time (Figure 4G,I). As with the astroglial cells cultured in 10% FBS, we did not detect any mitotic cells throughout the 5 days.

The milking of the brain protocol involves the compromise of the integrity of the ependymal cell layer at the level of the lateral ventricles, induced by the administration of neuraminidase [36,37]. Even though the areas of ependymal damage are later characterized by the formation of astroglial scars, in some cases, we also observed that some long-term, surviving ependymal cells were expressing markers of mitosis (Appendix A). To assess this phenomenon further, we used the ependymal cell cultures and exposed the cells to different doses of neuraminidase. We observed a reduction in the number of ependymal cells with the increasing dose of neuraminidase; although, this did not reach statistical significance (Figure 5A). Even though the presence of clusters remained unaltered (Figure 5B) at the higher dose, the morphology of ependymal cells was affected, with cells losing cell volume (Figure 5C). Notably, the presence of high doses of neuraminidase led to the appearance of mitotic ependymal cells, identified by the expression of phospho-histone 3, a protein expressed only during the M phase of the cell cycle. Mitotic activation was detected both in ciliated ependymal cells (Figure 5D) and in non-ciliated, elongated ependymal cells (Figure 5E). Since we found that ependymal cells, kept in differentiation conditions, could be induced to re-enter the cell cycle after ependyma-specific stress, we also assessed the behaviour of ependymal cells when cultured in the strongly pro-mitotic NSPC medium and we confirmed that the presence of FGF-2 and EGF led to the emergence of mitotic ependymal cells (Figure 5F).

## 4. Discussion

Astrocytes are the most abundant and the most multifaceted cell type of the CNS [45]. They play a role in regulating metabolic processes, the extracellular microenvironment, synaptic activity, and the response of the tissue to cell stress and degeneration. In addition, NSCs assume an astrocytic phenotype in order to survive in the postnatal brain [46] while astrocytes are the cells of choice in any attempt to induce glia-to-neuron conversion [47]. Therefore, the detailed investigation of the molecular processes that operate during the transition of a “mature” astrocyte towards a more progenitor-like identity is of high importance in any effort to harness the endogenous regenerative potential of the brain.

Here, we generated astrocytes by inducing NSCs isolated from postnatal mice to differentiate towards this fate by culturing them in 10% FBS. The astrocytes that were produced expressed a range of characteristic astroglial markers (*GFAP*, *S100β*, *Glast*, *Aldh1*, and *Sox9*), were not mitotic, were immunonegative for Sox2, and, upon exposure to further differentiation conditions, generated only astrocytes. Based on the above, this protocol allowed the generation, in high numbers and in standardized conditions, of a homogeneous population of mature astrocytes. It relieves the process from the introduction of confounding factors resulting from (i) the different levels of cell stress and damage occurring during the acute isolation of astrocytes from the brain [48], (ii) or during the application of complicated, typically involving serum, culture protocols [49], and (iii) the spatial heterogeneity of the brain’s astrocytes. It should be noted, however, that NSCs populating the SEZ are not of common embryonic origin; although, embryonic NSCs of ventral origin dominate in the formation of the niche [50]. The expression of *Pitx3* in the astrocytes we generated suggests a positional identity link with dopaminergic neurons of the substantia nigra in the midbrain [51] or with dopaminergic interneurons in the olfactory bulb, both being areas that receive SEZ-derived neurons [19,52]. A detailed investigation of the positional identity of these astrocytes and of any cells they might generate after reprogramming has to be performed in the future, taking also into account the effects of poorly defined serum factors included in the initial steps (e.g., pro-gliotic BMPs) [53]. This is why we then sought to explore the plasticity of these mature astrocytes, in terms of their de-differentiation capacity, in simple and clear ways by replacing the pro-astroglial medium with two different serum-free media. One that has been shown to favour the uncommitted NSC identity (referred to as NSC-stemness medium) [42], which includes a GSK3 and a TGF-β inhibitor, or another that did not contain the inhibitors but with the addition of FGF-2 in order to support cell survival and to promote lineage (referred to as NSC-lineage medium). The key results of this experiment were the massive re-appearance of the expression of Sox2, which was significantly higher in the NSC-stemness medium, and the re-emergence of mitotic cells, only in the NSC-stemness medium. We, therefore, were able to reverse astrocytes towards a progenitor identity but to different degrees. The highest reversal was achieved by inhibiting the GSK3 and TGFβ pathways, resulting in astrocytes that not only re-expressed the key progenitor factor Sox2 but also re-entered mitosis. To further assess the degree of de-differentiation we also investigated the expression of ID3, a marker of quiescent NSCs [43,54], and we found a ubiquitous lack of expression, in both conditions. This could be due either to a real limitation of the capacity of astrocytes to fully de-differentiate all the way back to a bona fide NSC or to a limitation of the medium to support the quiescent NSC (qNSC) identity [42,55].

To test for the latter, we isolated NSCs from the brains of live rats using the brain-milking method, with which cells are released in the CSF and subsequently collected via liquid biopsies, without the use of aggressive cell-dissociation protocols [36,37]. NSCs isolated with this method retain high levels of quiescence as they generate slowly growing colonies and exhibit limited self-renewal capacity, similar to endogenous SEZ NSCs [56,57]. When milking-derived cells were cultured in the NSC-stemness medium, we found a significant expansion of ID3+ cells, as compared to cells cultured in the NSC-lineage and the neurosphere medium, at the respective expense of the expression of EGFR, which characterises activated NSCs and their progeny [58,59]. Therefore, we demonstrated that the NSC-stemness medium not only can sustain the qNSC identity [42] but can also foster the expansion of these cells in vitro.

Lastly, we switched the culture medium of de-differentiated astrocytes to a typical differentiation-inducing, serum-free, growth-factor-free medium that does not bias towards specific identities or cell subtypes. Our results showed the emergence of oligodendrogenic potential, with astrogliogenesis remaining significantly lower in the astrocytes de-differentiated with the NSC-stemness medium. The observed lack of neurogenic activity, even in the most de-differentiated astrocytes, might be linked to the failure to generate ID3+ quiescent NSC since quiescence is necessary to unlock the full potential of stem cells [31,60,61]. Additional experimental work is necessary to clearly establish if the re-emergence of neurogenesis will require a deeper reversal of astrocytes towards the qNSC state and what the essential factors to facilitate this in culture are [55]. It should be, also, noted that neurogenesis might require longer time frames and is a much more complicated process, influenced by the developmental origin [62] and the exact location in the niche of the parent NSCs [18,19,58]. Nevertheless, the appearance of neuroblasts, initially expressing doublecortin and subsequently βIII tubulin, is typically observed within the first three days of differentiation [63].

Ependymal cells of the spinal cord are another glial type that has been shown to exhibit astrogliogenic and oligodendrogenic properties in vivo in response to injury [32]. Similar properties have not been reported for the brain’s ependyma, which cannot be regenerated after damage; instead, it is replaced by gliotic tissue [37,64] and ependymal loss can lead to hydrocephalus. Nevertheless, an immunohistochemical analysis of the lateral ventricles’ ependyma, at three months post-milking, revealed the presence of ependymal cells expressing the proliferation marker PCNA. Therefore, we set up ependymal cell cultures and we confirmed the presence of multiciliated, cuboid ependymal cells of different sizes. When we mimicked the stress that ependymal cells experience in vivo after the injection of neuraminidase in the lateral ventricles, by exposing ependymal cells isolated from the same area to neuraminidase, we detected the appearance of mitotic ependymal cells. Furthermore, we detected mitotic ependymal cells in the high presence of FGF-2 and EGF, both being NSC-supporting factors [43] that are present in the SEZ niche [21,65].

## 5. Conclusions

Overall, our data reinforce the hypothesis that astrocytes and ependymal cells retain the intrinsic potential to reverse toward a progenitor identity. They also show that different levels of de-differentiation can be achieved in vitro by manipulating the culture conditions; thus, they can be used in order to dissect the molecular signatures that govern the flow from the qNSC state to that of fully mature cells (such as neurons and oligodendrocytes) or of cell types with enhanced plasticity potential, such as astrocytes and ependymal cells. The aim of this work was to provide a well-characterized and simple platform for the investigation of the reprogramming potential of the brain’s astroglial and ependymal cells. It is intended to serve as a starting point that can be modified in order to investigate the activity of signalling pathways (e.g., the role of BMPs or of Notch that are known to play a role in the maintenance of the astroglial and neuronal identities [25,53,66,67] using serum-free, more tightly defined, conditions. In the absence of large NSC pools in the human brain, especially the ageing one [68], the capacity to reprogram in vivo astrocytes or ependymal cells remains a valid therapeutic target [14,26,47]. The employment of such cell assays, especially with the use of human neural progenitors of embryonic or iPSC origin, will accelerate the effort to devise reprogramming strategies using highly controllable small molecules or signalling factors, rather than the induced expression of transcription factors [69]. 

## Figures and Tables

**Figure 1 cells-13-00668-f001:**
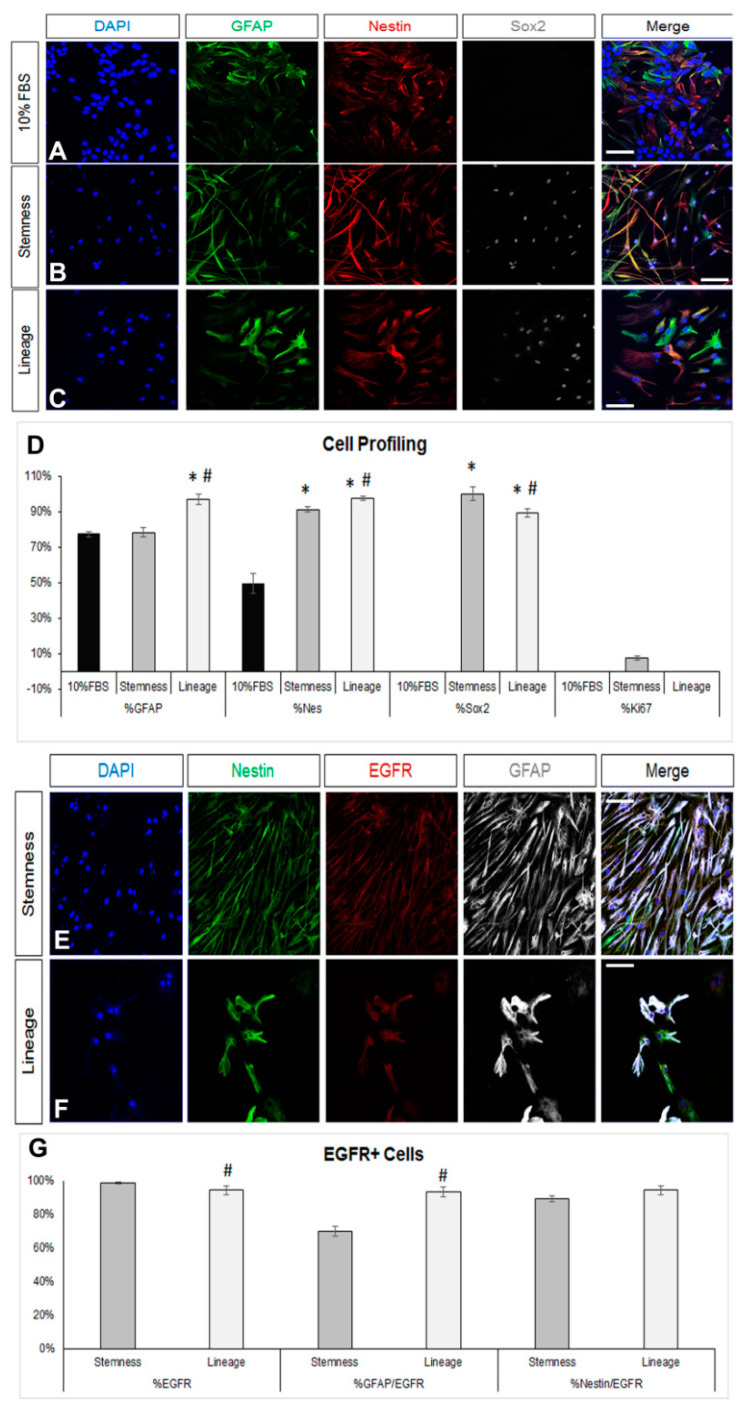
De-differentiation of astrocytes via modification of the culture medium. (**A**–**C**) Images of cells immunostained for GFAP (in green), nestin (in red), and Sox2 (in grey) immediately after culture in DMEM+ 10%FBS for 7 days (**A**) or after a further 7-day culture in two different de-differentiation media (**B**,**C**). (**D**) Graph showing the marker profile quantification of cells grown in the different media. (**E**,**F**) Comparative images of astrocytes immunostained for nestin (in green), EGFR (in red), and GFAP (in grey) after culture in the two de-differentiation media. (**G**) Graph showing the profile of EGFR+ cells after culture in the two de-differentiation media. [Scale bars: 20 μm; (**D**) *: *p* < 0.05 for comparisons to the 10%FBS medium and #: *p* < 0.05 for comparisons to the stemness medium; one-way ANOVA followed by post hoc analysis; (**G**) #: *p* < 0.05 for comparisons to the stemness medium; Student’s *t*-test].

**Figure 2 cells-13-00668-f002:**
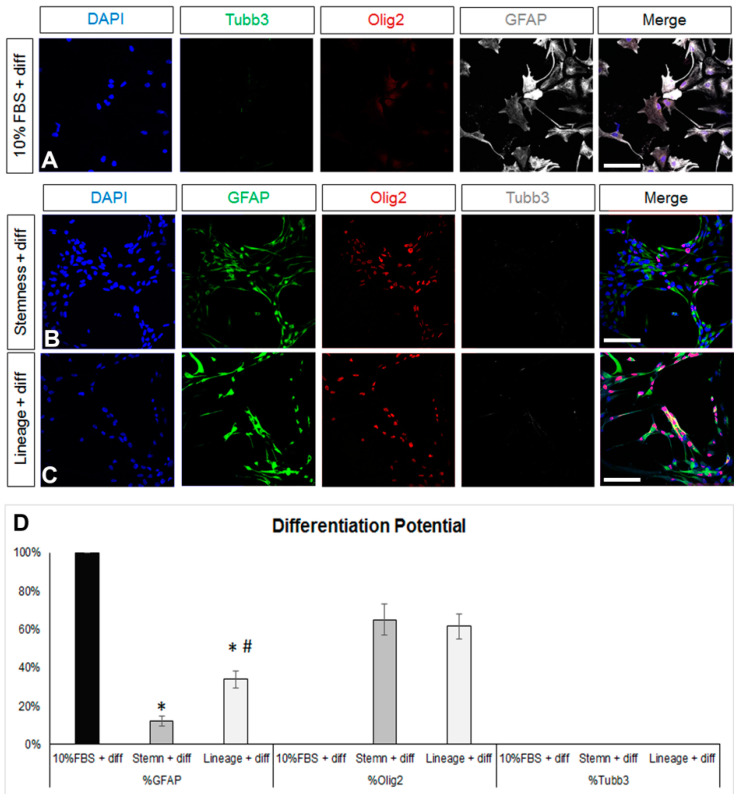
Investigation of the differentiation potential of mature and de-differentiated astrocytes. (**A**) Images of astrocytes generated after a 7-day culture in DMEM+ 10%FBS and then being differentiated for another 5 days in supplemented serum-free, growth-factor-free DMEM immunostained for β3-tubulin (Tubb3, to mark neurons, in green), Olig2 (to mark oligodendroglial lineage cells, in red), and GFAP (in grey). (**B**,**C**) Images of astrocytes de-differentiated for 7 days in two different media and then differentiated for another 5 days in supplemented serum-free, growth-factor-free DMEM immunostained for GFAP (in green), Olig2 (in red), and Tubb3 (in grey). (**D**) Graph showing the marker profile quantification of cells grown in the different media after 5 days of differentiation. [Scale bars: 20 μm; (**D**) *: *p* < 0.05 for comparisons to the 10%FBS medium and #: *p* < 0.05 for comparisons to the stemness medium; one-way ANOVA followed by post hoc analysis].

**Figure 3 cells-13-00668-f003:**
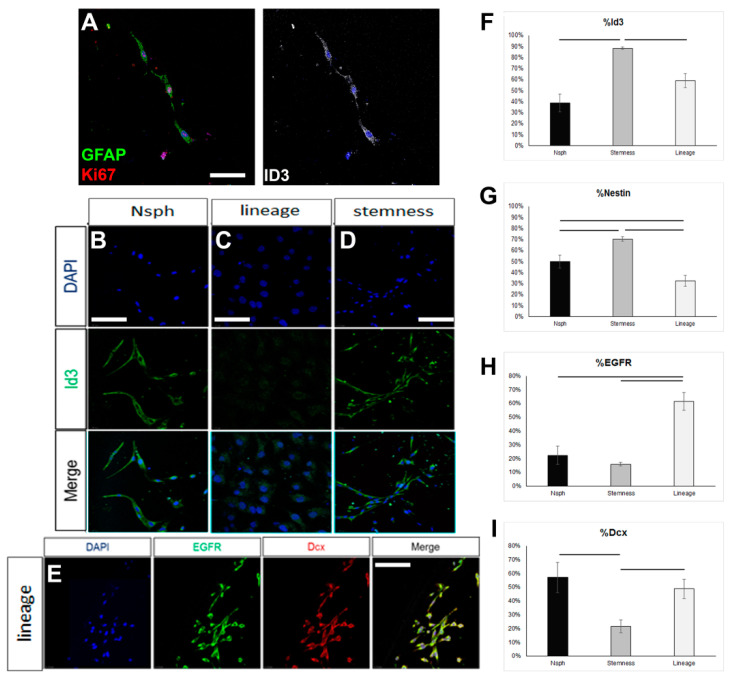
Investigation of the properties of rat NSCs cultured in different media. (**A**) Image of mitotic astrocytes (co-expressing GFAP, in green, and Ki67, in red), as well as ID3, in cell samples obtained via the milking of the brains of live rats and cultured for 7 days. (**B**–**D**) Images of ID3+ cells (in green) in samples obtained via the milking of the brains of live rats and cultured for 7 days in different media (Neurosphere/Nsph medium; NSC-lineage and NSC-stemness media). (**E**) Image of milking-derived cells immunostained for EGFR (in green) and Doublecortin (Dcx, in red) after culture for 7 days in the NSC-lineage progression medium. (**F**–**I**) Graphs showing the marker profile quantifications of milking-derived cells grown in the different media for 7 days. [Scale bars: 15 μm in (**A**) and 20 μm in (**B**–**E**); horizontal lines indicate difference (*p* < 0.05); one-way ANOVA followed by post hoc analysis].

**Figure 4 cells-13-00668-f004:**
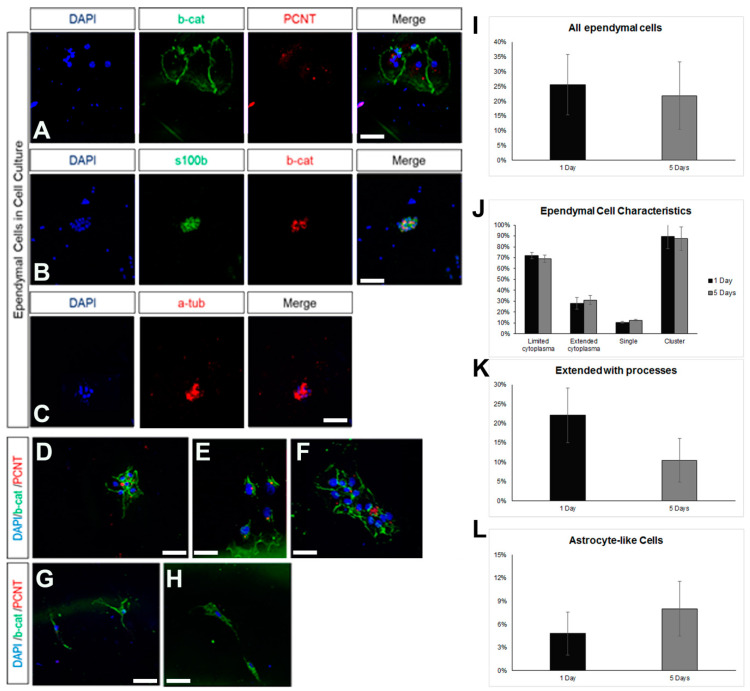
Characterization of ependymal cells in primary cell cultures. (**A**) Indicative image of ependymal cells (co-expressing beta-catenin, in green, and PCNT, in red) in cell samples obtained via the milking of the brains of live rats and cultured for 7 days. (**B**) Indicative image of ependymal cells (co-expressing s100beta, in green, and beta-catenin, in red). (**C**) Indicative image of ependymal cells expressing acetylated tubulin (in red). (**D**–**H**) Indicative images of ependymal cells (**D**–**F**) of different sizes and morphologies, as well as of astrocyte-like cells (**G**–**H**), expressing beta-catenin (in green) but not PCNT (in red). (**I**–**L**) Graphs showing the quantification of different types of cells after different time points in culture. [Scale bars: 15 μm].

**Figure 5 cells-13-00668-f005:**
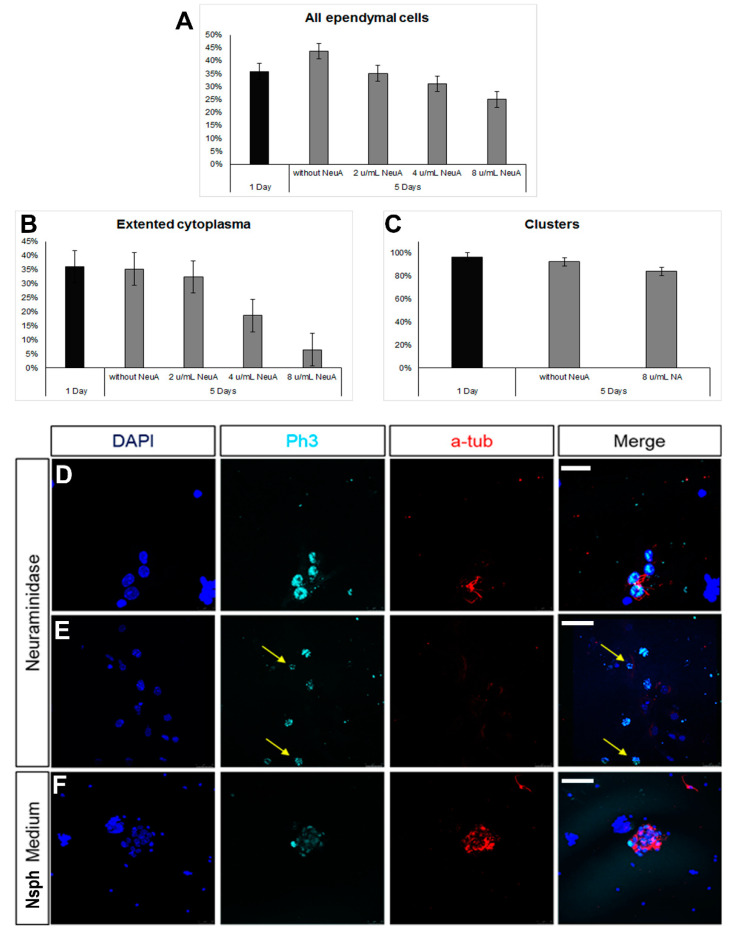
Emergence of mitotic ependymal cells in primary cultures. (**A**–**C**) Graphs showing numbers and morphological features of ependymal cells after exposure to different doses of neuraminidase. (**D**–**F**) Images of cells expressing phospho-histone3 (in cyan, to mark cells in the M phase). Some of these cells were ependymal cells (expressing acetylated tubulin, in red) and some were not (examples shown with yellow arrows). [Scale bars: 10 μm in (**D**), 20 μm in (**E**,**F**)].

## Data Availability

All raw data supporting the conclusions of this article will be made available by the authors on request.

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
