# Peer review of "Reversal of Postnatal Brain Astrocytes and Ependymal Cells towards a Progenitor Phenotype in Culture"

_cells, 2024, doi:10.3390/cells13080668_

Round 1

Reviewer 1 Report

Comments and Suggestions for Authors

The hypothesis of the article is of high importance for understanding the molecular mechanisms that regulates the transition from a differentiated cell state towards a progenitor identity and vise versa. 

The introduction and the discussion are well structured. The details of the protocols are provided in materials and methods, ensuring reproducibility.

However, the way that the experiments are performed and presented is not very clear and cannot support the conclusions drawn. Lineage tracing experiments are necessary to address the points raised in the manuscript.

Some suggestions to make the text easy to follow are the following:

-A graph depicting the experimental procedure in the figures would be helpful to allow the reader follow the text  and understand easily the results of the paper.

-A more detailed description of the protocol and a consistent naming of different media used would also be helpful in the beginning of results, in order to avoid confusion between the different experimental conditions. For example, astrogliogenic medium and de-differentiation medium (Line 287 ), multilineage pro-differentiation medium (Lines 250 and 272 ), Nsph Medium (figure 4B)  have not been referred previously in the text with these names- making it confusing which medium is used in each step.

-The same name for tubb3 (βIII tubulin and b3 tubulin) should be used in text and figures.

-The word lineage is repeated (Line 151)

Some comments regarding the experimental content 

-Figure 1 : is demonstrated in the introduction but is not very relevant to the rest of the article’s content.

-The population of astrocytes used in the first part of the paper is Neural Stem Cells that have  astroglial morphology. The cells are isolated from the SVZ of adult mice and are differentiated towards astrocytes and their de-differetniation capacity is tested with different media. A more clear set up could be to isolate directly astrocytes from the cortex of mice and see the plasticity of these cells for more robust results. 

-Figure 2 and lines 255-269 : Most markers are common- it is not clearly stated what these markers show and which is the difference between the two panels of images. In(2D) Ki67 is shown as a percentage in the graph. A representative image should be included in the figure along with the other markers.

-More astroglial markers (Glast, Aldh1 and Sox9) are mentioned in the text and are not shown in images. Representative images should be provided to support the results extracted from the experiments.

-The importance of Pitx3 expression in the de-differentiating of astrocytes in the expense of other positional markers (line 247) is not clarified.

-Figure 5 :  

-(5A) in figure legend ID3 is mentioned as a marker that has been used in samples obtained via brain milking_ no image is provided and there is no profound reason to mention that this marker is used as an indicative marker in ependymal cell cultures

-(5A-C)Higher magnification images should be taken to provide evidence that cells in the culture are ependymal cells- it should be clear that there are multiple basal bodies  (accumulation of PCNT) and that there are multiple cilia (stained with acetylated tubulin)

-(5D-F) The b-catenin staining is not optimal and it is not clear how the classification of the types of ependymal cells is done- most of them do not seem to have multiple basal bodies-more representative photos are needed to show what authors describe.

 -Figure 6: 

-As mentioned before, images are not very convincing- higher magnification images should be provided to show clearly the cilia. 

-(6D) there is no scale bar and the images seem to be of different magnification than E and F .

-(6E) is described as two different conditions in the text (line 388 and 393).

-(6F) it is not clear what full medium is and is not described in the text.

- In line 398 is mentioned that some cells expressing PH3 are ependymal cells. It would be important to count the percentage of the ependymal cells in the cultures that are PH3+ after high doses of neuroaminidase  in order to identify the percentage of ependymal cells that are in mitotic state as described. Ιn addition the PH3+ cell populations that are not ependymal cells should also be quantified in full medium and after neuroaminidase addition.

 Overall the idea of the article is very interesting but the data provided are not sufficient to support the conclusions that are extracted. There are many points that need to be clarified in the results and in figures in order to re-condider the data. 

Author Response

Please see the attachment, that includes collectively our responses to all reviewers.

Reviewer 2 Report

Comments and Suggestions for Authors

The manuscript by Kakogiannis et al presents an interesting methodological study. The authors studied the features of de-differentiation and re-differentiation of astrocytes. The ability of differentiated ependymal cells to undergo mitosis was also examined.

The section of the article on ependymal cells is quite interesting. There are few studies of ependyma in vitro, and this work is interesting and novel.

However, the part about astrocytes raises questions.

Introduction – more relevant references should be provided. To date, there are many works devoted to the differentiation and dedifferentiation of cells in culture. The authors refer mainly to fairly old articles published in 2002-2013.

Materials and methods

What are B6CBAC mice? Are these hybrids or a line? Why did the authors take these unusual mice?

Why did the authors obtain astrocytes by NSC differentiation? In the discussion section, the authors mention that this method has an advantage over obtaining primary cultures of mature astrocytes, but do not explain which one. How do astrocytes obtained using the method of the authors differ from primary astrocytic cultures of mature astrocytes?

In addition, the SVZ contains not only progenitor cells but also differentiated astrocytes. Why do the authors believe that their astrocytes were obtained precisely by differentiation from precursors? What did the cultures look like and what markers did the cells express before differentiation?

Author Response

(The authors gave the same response as above.)

Reviewer 3 Report

Comments and Suggestions for Authors

Introduction

Figure 1: How was the procedure and staining performed? Was the figure borrowed from alternate source, if so, the images must be included with permission from the author/publisher and duly mentioned.

Method

The original article refers to “Milking” (McClenahan et al.,  Stem Cell Reports . 2021;16(10):2534-2547), and  the authors should refer the same. Surprisingly, the article referred by the authors does not exist (J. Vis. Exp., e65308).

Results

There are several discrepancies regarding the method used for the isolation of the stem cells.

1.       How were the cells characterized after isolation procedure? How did the authors exclude the Oligodendrocyte progenitor cells from their preparation? Based on my observation, bulk cell population, and not just NPSCs were present in the study, which may affect the results from the study.  Even the article referred by the authors in the article (un-identifiable) shows OPSCs in their preparation.

2.       In section 3.1, lines 235-239: the authors mention that cells from Subependymal zone stem cell nitches were used, however,the based on the method the astrocytes were isolated using neuraminidase activity.

3.       Figure 2B: Missing EGFR marker data for FBS alone in the figure or the supplement.

4.       Lines 246-249: the expression of these markers must be included either in the article or in the supplementary.

5.       Figure 2D: Images from Ki67 are missing. Based on the Fig 2D, there were cells positive for this marker which must be shown. Also the graphs 2G does not represent the total number of cells counted. Based on the number of Dapi + cells, the cells were not evenly distributed. Moreover, how was the counting performed?

6.       Lines 251-254: the authors mention no Olig2 or TuB3 markers, but based on figure 3, the cells are clearly positive for Olig2. Moreover, there is no uniformity in the figure profile.

The authors must submit the all original uncropped images used in the manuscript for better understanding.

Lines 387-393: How did the author distinguish the ciliated and nonciliated ependymal cells? There is no morphological difference between the cells in the presented images. Moreover, the conclusion presented the authors is incorrect. Based on the figures D-F, the maximum Ph3 expression is observed only when neuraminidase is present alone, but not in the full medium. I also think, the images are not of the same resolution, this must be rectified.

7.       Figure 4: Missing  EGFR images considering the cells show positive expression of this marker.

Ethical approval: Details regarding the ethical approval is missing in the article. 

Author Response

(The authors gave the same response as above.)

Reviewer 4 Report

Comments and Suggestions for Authors

These studies are based on previous findings suggesting that at least in vivo, astrocytes and ependymal cells could retain a high capacity to reverse to a progenitor state of differentiation. 

The authors now report that under different in vitro culture conditions, both cell types can indeed be induced to reverse towards a progenitor-like identity and different levels of de-differentiation can result from the use of different cell culture medium conditions. The results are very interesting and potentially important for the design of CNS regeneration treatments.

There are some points that could strengthen the manuscript and analysis of results.

- In the Introduction section: References are not up to date. At the beginning of the introduction, when talking about astrocytes, the statement "Until very recently they were considered as a relatively homogeneous population of supporting cells......" represent an idea that is not "recent" anymore.  

- It is not clear if Figure 1 represents results from the authors or on the other hand, it belongs to any of the cited references in the corresponding text in the Introduction. If these are new results, it should be presented in the results section.  Otherwise, Figure 1 should be eliminated and only its description should be included with the cited references in the Introduction. 

- Under Results: Very important for the interpretation and validity of the results is the assurance of no bleeding of the green fluorescence when looking under the red channel. Controls figures should be provided for green staining alone observed under the red channel.

- Methods: Under Astrocyte generation and culture: CHIR-99021 should be described as "GSK-3 inhibitor CHIR-99021". The same applies for A83-01, it should be described as the ALK4/5/7 or TGF-beta/Smad inhibitor A83-01. Also, their concentrations are described as "um", instead of presumably uM?

Author Response

Please see attached with responses to all reviewers.

Reviewer 5 Report

Comments and Suggestions for Authors

The subject of this MS is exciting. The nature of the neural stem and progenitor cells in the adult brain and the plasticity of the postmitotic cells is a hot topic and still needs clarification in the field. Rejuvenation or reversal of the postmitotic cells in the brain is also a hot topic.

However, it is confusing to claim that  “Astrocytes are widely scattered in the brain exerting multiple functions and, also including Neural Stem Cells (NSCs), clustering within stem cell niches, and parenchymal cells that act as multipotent progenitors in response to injury.” 

The patterning process is of tremendous importance in the central nervous system development and regeneration, which confers unique profiles to the stem/progenitor cells, defining their fate. Here is the main problem of this MS. It does not address the pattering/subregions of the CNS cells.  Are they from the subventricular zone of the lateral ventricles from the dorsal or the ventral part? Which location of the ventral part? The same is valid for the ependymal cells. It is also unclear what regions form the “Milky Brain,” what it contains, and how significant it is for this study.

The neurogenic capacity of the NSCs should be validated in particular conditions not included in the presented methods.  Neurogenesis cannot be excluded from some specific regions of the SVZ of the lateral ventricles.

The mechanisms to control the reversal of neural cells from a postmitotic to a proliferative state are far from being clarified in this paper, but some preliminary data are encouraging. It needs a precise experimental setting, additional markers, and better-defined conditions.

Comments on the Quality of English Language

English is good, minor editing is required.

Author Response

(The authors gave the same response as above.)

Round 2

Reviewer 1 Report

Comments and Suggestions for Authors

The authors did not reply to our major comments.

In my first review I have stated that the manuscript is rejected mainly because the manuscript makes conclusion without being supported from experimental evidence.

The experiments provided did not changed my opinion.

Author Response

The authors did not reply to our major comments.

In my first review I have stated that the manuscript is rejected mainly because the manuscript makes conclusion without being supported from experimental evidence.

The experiments provided did not changed my opinion.

New response:  It is unfortunate that the reviewer felt that their comments were not addressed successfully and did not come back with more focused points.

Reviewer 3 Report

Comments and Suggestions for Authors

The authors have addressed the issues raised by this reviewer.

Minor comment:

The article  on the isolation procedures (in consideration) must be cited appropriately.

Comments on the Quality of English Language

Requires minor revision.

Author Response

Minor comment:
The article on the isolation procedures (in consideration) must be cited appropriately.

New response: We thank the reviewers for their positive assessment of the revised manuscript.
The cited paper is now published, and all the details have been correctly included in the literature list.

Reviewer 4 Report

Comments and Suggestions for Authors

Previous concerns were appropriately addressed.

Author Response

Previous concerns were appropriately addressed.

New response: We thank the reviewers for their positive assessment of the revised manuscript.

Reviewer 5 Report

Comments and Suggestions for Authors

The revised MS is significantly improved. However, not all my comments were addressed, and some confusion should still be clarified.

Previous comment 1: It is confusing the claim that “Astrocytes are widely scattered in the brain exerting multiple functions and, also including Neural Stem Cells (NSCs), clustering within stem cell niches, and parenchymal cells that act as multipotent progenitors in response to injury.”

Response 1: The abstract has been revised and the specific part now reads as follows: “Astrocytes are widely scattered in the brain where they exert multiple functions. Additionally, Neural Stem Cells (NSCs), clustered within stem cell niches, and latent parenchymal progenitors that become activated in response to injury are also astrocytes.”

New comment 1: Please revise the abstract and consider it as it should contain only the specific results obtained by the experiments (in a very condensed form but fully explanatory). Please clarify that the astrocytes are not neural stem cells, but they can have similar morphology and share some markers in the restricted areas of the adult brain.

Previous comment 2: The patterning process is of tremendous importance in the central nervous system development and regeneration, which confers unique profiles to the stem/progenitor cells, defining their fate. Here is the main problem of this MS. It does not address the pattering/subregions of the CNS cells. Are they from the subventricular zone of the lateral ventricles from the dorsal or the ventral part? Which location of the ventral part? The same is valid for the ependymal cells. It is also unclear what regions form the “Milky Brain,” what it contains, and how significant it is for this study.

Response 2: The patterning issues raised by the reviewer are of great importance in terms of the biology of NSCs and generally of the brain’s cells. However, they do not confound this level of experimental work because we are comparing the effects of different culture media on the same biological samples. The possible existence of different properties in cells having been isolated from different areas of the brain (e.g. NSCs isolated from the hippocampus vs those isolated from the SEZ, or ependymal cells isolated from the central canal vs those isolated from the lateral ventricles) is an interesting follow-up line of work, but remains out of the scope of this founding work. We (and others) routinely isolate NSCs by dissecting the whole lateral walls of the lateral ventricles (where the main corpus of the SEZs are located) of both hemispheres. Indeed, previous experimental work (e.g. from Fiona Doetsch’s lab) has indicated that different subdomains of the SEZ harbor NSCs with differential properties; this is something that cannot be easily addressed with the approach used (microdissections) in this study and necessitates the parallel use of (transgenic or virus-mediated) labelling strategies. We revised the respective Materials and Methods section to read as follows: “To culture postnatal brain neural stem and progenitor cells in the form of neurospheres, the whole lateral walls of the lateral ventricles (where the SEZs are located) of adult mice were dissected under a stereoscope, dissociated…”. Accordingly, we revised the section describing the isolation of ependymal cells: “Ependymal cells were obtained by dissecting the periventricular areas, around the lateral ventricles, of adult mice, under a stereoscope.” Regarding “milking”, the coordinates in which cells are released from the tissue in the CSF have now been added to the Materials and Methods text, that reads as: “Briefly, rats were stereotaxically injected (coordinates: anterioposterior=0.3mm, lateral=+1.2mm, depth=3,5mm) with 2μl of a release cocktail, consisting of Clostridium perfringens neuraminidase…” We have previously reported that by targeting these coordinates we can collect cells originating from the SEZs (McClenahan et al., 2021, Stem Cell Reports; Dimitrakopoulos et al., 2024, JoVE). The “milking of the brain method” allows the direct isolation of NSCs without the necessity to kill the animal to dissociate the brain tissue. We have shown that in this way NSCs retain in higher fidelity their in vivo properties. Here, we use NSCs isolated by milking to assess, in the most relevant cell population, whether the NSC-stemness medium can foster quiescence in culture.

New comment 2: Thank you for clarifying from which area the cells used in this paper are obtained. This should be mentioned in the abstract and maybe in the title. The characterisation of the progenitors should contain telencephalic markers and markers of different regions of the LV wall. As mentioned in the answer, many previous experiments (e.g. from Fiona Doetsch’s lab) indicated different subdomains of the telencephalic SEZ (LV wall) ) and the different fates of the  NSCs they contain. It is essential to check the presence of the main telencephalic patterning markers in the cell populations before and after the treatments.

Previous comment 3: The neurogenic capacity of the NSCs should be validated in particular conditions not included in the presented methods. Neurogenesis cannot be excluded from some specific regions of the SVZ of the lateral ventricles.

 Response 3: In this study we take for certain that the SEZ-derived NSCs have a neurogenic potential and we confirm that is the case when they are cultured in all the NSC-specific media assessed (the typical neurosphere medium, the NSC-lineage and the NSC-stemness media), in which cells of neuronal commitment (expressing doublecortin) are present (revised Figure 3E and I). What we highlight is that the de-differentiated astrocytes do not exhibit neurogenic potential, irrespective of culture conditions.

New comment 3. It is very important to clarify what kind of neurons are generated from these neural stem progenitor cells in the proposed conditions; only doublecortin staining is not enough for this level (of publication).

Previous comment 4: The mechanisms to control the reversal of neural cells from a postmitotic to a proliferative state are far from being clarified in this paper, but some preliminary data are encouraging. It needs a precise experimental setting, additional markers, and better-defined conditions.

Response 4: We thank the reviewer for kindly acknowledging the value of our results and we, obviously, realize the many limitations and challenges of this type of experimental work and the necessity for further follow-up.

New comment 4. While I acknowledge the value of the MS, I consider that the results of the experiments are not sufficiently valorised and discussed, e.g., in connection with the differentiation pathways in fetal life and adults.  The use of the fetal calf serum as an undefined condition represents a minus, and it should be replaced or discussed in the context of the mechanistic pathways of differentiation or proliferation.  It should also be discussed why this approach is relevant and how it could be translated into human studies. 

Comments on the Quality of English Language

Minor editing of the English language is required.

Round 3

Reviewer 5 Report

Comments and Suggestions for Authors

In the revised version of the MS and in the cover letter with the answers, the authors addressed several of my previous comments. However, several comments are not correctly addressed and there are several lines in the MS that contain scientific mistakes (marked in yellow in the attached document). The lines 238-240 contain incorrect statements:

"Moreover, we looked for the expression of positional 238 brain markers, such as Tbr1, Pitx3 and Gbx2 (marking forebrain, midbrain and anterior 239 brain identity, respectively) and only Pitx3 was found to be expressed (data not shown)".

Tbr1 is not a forebrain marker, but a marker of several glutamatergic projection neurons in several brain areas during development, including the neocortex. Pitx3 is not a midbrain marker, being expressed in several areas of the brain and neuronal types, including the olphactory bulb, where the adult neurogenesis takes place in mice. Gbx2 is definitely not an anterior marker, but a posterior one. 

The new answers in the cover letter to comment 2 " we used Tbr1 as a typical forebrain marker and we found it was not expressed in our cells, strongly suggesting absence of forebrain identity" is again incorrect. "Thus, the dominance of a midbrain identity might not be a total surprise" is not justified.

The incorrect explanations in the cover letter should be reconsidered "It should be noted that the generated astrocytes exibited a midbrain identity, with no expression of Tbr1, a typical forebrain marker. This could be explained by at least two ways; firstly, it might reflect the known dominance of embryonic NSCs of ventral origin in the formation of the SEZ [51], which could characterise their progeny, too. Secondly, it might be the homogenizing result of poorly defined factors included in the serum (e.g. pro-gliotic BMPs)[52]."

The correct explanation is the forebrain neurogenic areas in adult brain contain cells expressing Pitx3, but not Tbr1 expressing neurons (glutamatergic), in line with many papers cited in the references.  The authors found expression of Pitx3, but not Tbr1 and Gbx2. Their attribution of this expression profile is incorrect.

The whole part of the paper including the lines 238-240 (and in the methods the corresponding lines 204-212) should be deleted or a revised, correct explanation for the provided findings should be provided.
